# Sustainable Population Size at the County Level under Limited Development Policy Constraints: Case Study of the Xihaigu Mountain Area, Northwest China

**DOI:** 10.3390/ijerph19159560

**Published:** 2022-08-03

**Authors:** Xueli Chen, Yongyong Song, Xingang Fan, Jing Ma

**Affiliations:** 1School of Economics and Management, Ningxia University, Yinchuan 750021, China; 12020130066@stu.nxu.edu.cn (X.C.); maj30@nxu.edu.cn (J.M.); 2Research Institute of Western China Development, Ningxia University, Yinchuan 750021, China; 3School of Geography and Tourism, Shaanxi Normal University, Xi’an 710119, China; syy2016@snnu.edu.cn

**Keywords:** population size, environmental quality, population carrying capacity, sustainable development, limited development ecological zone, Xihaigu Mountain area

## Abstract

Understanding the extent to which demographic changes may affect the prospects of sustainable development is a priority for both academics and policy makers. Accordingly, we attempted to explore the population growth limit of the Xihaigu Mountain area in China. To analyze the optimum population at the county level, the relationship curve between population size and environmental quality was fitted using panel data (2009–2018). The sustainable population size of each county was determined by integrating the population carrying capacity of land resources and optimum population. The results show that the relationship between regional population size and environmental quality conforms to an inverted N-shaped curve. To maintain sustainable development, the population size of Tongxin, Xiji, and Haiyuan should be in the range of 320,800–379,800, 315,800–416,900, and 333,500–416,900, respectively. The current population size of other counties should be maintained, and their surplus construction lands are acceptable to be used for environmentally friendly industries rather than population expansion. We demonstrated a practical approach to calculate a dynamic range of population size under the dual constraints of resource and environment, which overcomes the shortcoming of only considering the maximum carrying capacity to a limited extent. We also identified the population boundary in a “steady-state economy” and quantified planetary boundaries of population in Xihaigu Mountain area using a dynamic sustainable population size. The findings provide decision-making references for the local government.

## 1. Introduction

Understanding the factors that regulate the size of human populations is crucial to the development of an ecologically sustainable society derived from Malthus’s “principle of population” and the concept of “steady-state economy” proposed by Herman E. Daly [1]. Similar to the concepts of “ecological threshold”, “planetary boundaries”, and “resource and environmental carrying capacity”, sustainable population size is concerned with the identification of the boundary of sustainable economic and social development, which has been a controversial point within the framework of the Sustainable Development Goals (SDGs) [2,3,4]. In other words, sustainable population size is another interpretation of ecological economics with respect to resource and environmental carrying capacity. Sustainable population size is the upper limit of the number of people that the resources and environment of a given region (socioeconomic system) can carry without destroying the natural ecological environment and ecosystems [5]. Sustainable population size directly measures and regulates the output of “stressors” of human activities [6]. Given the recent trends in population growth and predictions regarding its evolution in the coming decades, understanding the extent to which demographic changes may affect the prospects of sustainable development is a priority for both academics and policymakers. In its report [7], the Future Earth Transition Team noted that, “Some regions, people and ecosystems are more vulnerable than others to global environmental change because they are located in places where changes are the most extreme, where biodiversity is the greatest, where populations are especially sensitive, concentrated or poorer, or where parts of the Earth system or local ecological systems are closer to thresholds”. In addition, the problem of resource and environmental overload in underdeveloped areas in China is becoming increasingly serious [8,9]. Accordingly, China implemented major function-oriented zoning (MFOZ) in 2011 [10]. One of the development goals is to reduce the total population and improve the population quality, i.e., enhancing human capital. A moderate size of human population can promote major functions, whereas an inappropriate size will inhibit them [11,12]. In this research, we take Xihaigu Mountain area as an example to explore the sustainable population size of restricted development ecological zones, i.e., areas with fragile ecological environments in China. According to MFOZ, Xihaigu Mountain area is included in limited development ecological zones, which are identified as areas with relatively low resource and environmental carrying capacity, as well as economic and demographic conditions unfavorable to large-scale development [13]. Xihaigu Mountain area has long been a key area of concern due to its unique characteristics, which are reflected in the following four aspects: (1) The population is growing rapidly. Xihaigu Mountain area is known for poverty and a fragile ecological environment caused by overpopulation [14]. The research of Mi and Zhou [15] and Li [16] showed that the population of Xihaigu Mountain area was growing too rapidly, especially as a minority area with relatively liberal fertility policies, leading to its rapid population growth. (2) Xihaigu Mountain area is located in the arid zone and is ecologically fragile. The excessive proliferation of population has exceeded the effective capacity of rigid land resources [17]. (3) The economy is underdeveloped. Due to the poor natural conditions, it was identified as an area unsuitable for human survival by the United Nations Food and Development Programme in 1972, and it is among the areas with the deepest poverty, the worst ecological conditions, and the most significant development difficulty among the six concentrated and contiguous areas with special difficulties in China [18]. (4) Xihaigu Mountain area pioneered China’s planned, organized, and large-scale, development-oriented poverty alleviation [19]. Since 1983, in order to alleviate poverty and restore the ecosystem, the government has successively implemented several resettlement projects in Xihaigu Mountain area, moving the surplus population north to the Yellow River diversion irrigation area [20]. For a long time, the contradiction between population, resources, environment, and socioeconomic development in the region has continued, and there is a tendency of a “PPE strange circle” between poverty, population, and ecological environment, which restricts the sustainable development of the region [21]. Above all, there is a contradiction between the protection of the ecological environment and the development of population and economy, and how to solve this problem has become a common concern of governments at all levels. Therefore, it is a necessary proposition to study the problem of “how large-scale development can be sustainable” in Xihaigu Mountain area.

As a core indicator of sustainable development, the population carrying capacity of a county, an area, or a region has always been a complex discussion. Nevertheless, under the premise of the environmental bottom line and resource upper limit and under different development stages and resource and environmental endowments, the regional population carrying capacity can be objectively determined [22]. In addition, due to the dynamic changes in the relationship between population resource consumption, environmental pollution, and economic development level, there is uncertainty with respect to the static population carrying capacity in a given space, reflecting an important trend from static analysis to dynamic prediction [23]. Therefore, using the limited development ecological zones of the Xihaigu Mountain area as a case study, we attempted to demonstrate a practical approach to calculate a dynamic range of population size under the dual constraints of resource and environment to overcome the shortcoming of only considering the maximum carrying size.

In this study, we identified the population boundary in a “steady-state economy” and quantified planetary boundaries of population in Xihaigu Mountain area using a dynamic sustainable population size. The specific objectives of this study were as follows: (1) to investigate the optimum population and current environmental conditions of counties in the Xihaigu Mountain area from the perspective of environmental constraints by developing a population-size and environment-quality model; (2) to calculate the population carrying capacity of each county and analyze their constraints of land resources from the perspective of land resource constraints; and (3) to evaluate the dynamic range of sustainable population size in each county. We hope that the findings of this study will provide references for decision making with respect to population size regulation to guide future development of ecologically fragile and underdeveloped areas.

The remainder of this paper is organized as follows. In the Section 2, we summarize the specific environmental and economic issues raised in the Section 1 and expound on the theoretical significance of the research. In the Section 3, we provide a general overview of the study area. In the Section 4, we present our methods and data. In the Section 5, we report the empirical results, analyze the population size status and possible causes under the constraints of environmental quality and land resources in all counties of study area, and provide the dynamic range of population size that should be maintained in order to achieve sustainable development. The results are further discussed in the Section 6. Lastly, our conclusions, policy implications, limitations, and further research suggestions are described in the Section 7.

## 2. Conceptual Framework

Steady-state economics emerged in the 1970s. As a new paradigm of modern economics, steady-state economics takes “development without growth” as its pre-analytic vision [24]. Traditional economics is limited to the study of the allocation of relatively scarce production factors and the distribution of income, and it is incorrect to assert that high-speed economic growth can continue indefinitely because excessive growth will accelerate the depletion of raw materials and aggravate environmental pollution, making economic growth lose its material basis. Moreover, although the utilization efficiency of a given resource will increase due to technological progress, the consumption rate of this resource tends to increase instead [25]. That is to say, only relying on technological progress is not enough to solve resource and environmental problems, and it is necessary to restrain endless growth and consumer demand. The steady-state economy (SSE) is clearly defined by Herman E. Daly as “an economy with constant population and constant stock of capital, maintained by a low rate of throughput that is within the regenerative and assimilative capacities of the ecosystem” [1]. To ensure efficient loading without capsizing, the weight of the cargo on board needs to be kept within the “Plimsoll Line” [26]. Likewise, to achieve sustainable economic development, a series of SSE policies needs to be adopted to keep the environmental effects of output within the carrying capacity, such as by limiting population within the sustainable population size range.

One important potential mechanism of human population regulation is the negative relationship between population size and per capita resource availability, which generates density-dependent or “Malthusian” growth [27]. Even if the renewal capacity of natural resources is unbounded, a sustainable path may not always be available; this depends on the difference between the stationary fertility rate and mortality rate [28], i.e., population size. In recent years, most studies on sustainable population size have been focused on the relationship between various factors, such as population growth and depletion of resources [29,30], global warming and urban population growth [28,31], population growth caused by migration and its resources and social effects [32], and population growth and political equality [33]. In addition, the environment is an essential “factor of production” and source of important welfare services to people [28], and many studies have focused on the interaction between global or regional population size and the environment [34,35,36,37,38], although few studies have been conducted on the effect of regulation of population size on sustainable local environmental development [39]. When the population size exceeds the carrying capacity of local resources and the environment over a long period, the local boundary may evolve into a global planetary boundary that will eventually be broken. Therefore, the human population in each country and the sustainability of its size should be discussed [40].

A sustainable population size provides a policy basis for limiting the population to achieve sustainable development, and its essence is the maximum population size under the overall ecological constraints. Therefore, in this paper, the optimal population size, population carrying capacity, and the current population size are superimposed to quantify the dynamic range of the sustainable population size. Among them, the optimal population size is derived from the environmental population inflection point, that is, the population size at which the environmental quality changes from increasing to decreasing, whereas the environmental quality index is constructed from three perspectives: the degree of environmental pollution, the foundation of the economic system, and the energy consumption intensity of economic activities. Furthermore, land resource were also taken into consideration as an important rigid constraint in designing this study framework.

## 3. Study Area

In terms of administrative divisions, the Ningxia Hui Autonomous Region is a provincial-level unit in China with 22 counties (districts) under its jurisdiction, including the 7 counties of Xiji, Longde, Jingyuan, Pengyang, Haiyuan, Yanchi, and Tongxin (Figure 1). Under the strategy of China’s MFOZ, these seven counties are included in the “restricted development ecological zone” due to their fragile ecological environment. Geographically speaking, these seven counties are located in the south of Ningxia in the famous Xihaigu Mountain area. In summary, our study area comprises seven counties (in the sense of administrative division), the Ningxia restricted development ecological zone (in the sense of policy), and the Xihaigu Mountain area (in the sense of geographical location).

The seven counties of the study area include 87 towns and townships, covering an area of 3.04 × 10^4^ km^2^ and accounting for 45.74% of Ningxia. As of 2018, the total population of the seven counties was 1,710,600, accounting for 24.86% of the total population of Ningxia. The seven counties represent the major habitation of the Hui people in China. Specifically, the population of the Hui people is approximately 956,800, accounting for 55.93% of the seven counties and 38.04% of Ningxia. The regional GDP is CNY 37.906 billion, accounting for only 10.80% of the GDP of Ningxia. According to *Ningxia Statistical Yearbook (2010–2019)*, the total population of the study area increased from 123,800 (1980) to 171,100 (2018), an increase of 38.2%, with the population size reaching 193,100 in 2006 as a result of policy control. The average natural population growth rate of the study area in 2009–2018 was 10.75‰, which is obviously higher than the average level of 8.73‰ in Ningxia.

The region has a typical temperate continental climate and is located in the farming-pastoral zone of northern China in the transition from the hilly area of the Loess Plateau to the desert steppe region. The annual precipitation is approximately 200–600 mm. Specifically, 63% of the desert steppe areas experience less than 200 mm of precipitation annually. However, evaporation in this region ranges from 1300 to 2900 mm. In this region, water resources are scarce, and the ecology is fragile.

## 4. Methods and Data

### 4.1. Research Methods

In this section, we mainly focus on three aspects of sustainable population size calculation. First, by constructing a population-size and environment-quality model, we calculated the average optimum population of the seven counties under the elastic constraint of ecological environment. Second, we calculated the actual population carrying capacity of each county by taking land resources as a hard constraint. Third, the optimal population and population carrying capacity were superimposed on the current level to determine the dynamic range of sustainable population size.

#### 4.1.1. Population-Size and Environment-Quality Model

(1).Population-Size and Environment-Quality Analytical Framework

According to the environmental Kuznets curve (EKC), environmental deterioration will first be aggravated and then improve with increased per capita income, presenting an inverted U-shaped curve (Figure 2a). However, the application of EKC is likely to be challenging in the long term because once clean technologies are fully utilized, even if the per capita income continues to increase, there is no more potential to be tapped. Subsequently, the increase in the cost of pollution reduction could lead to an increase in pollution levels, which may present as an N-shaped curve [41] (Figure 2b). Most of the results using a comprehensive index are shown as N-shaped curves [42,43,44]. Furthermore, changes in population size and per capita income are consistent (Figure 2c). As West pointed out in his book, per capita income grows at an exponential rate of 1.15 as the urban population grows, showing a kind of “superlinear scaling” [5], known as “increasing returns to scale” in economics [45,46]. “The larger the city, the more innovative the way to create social capital, and the greater the per capita ownership, production and consumption of ordinary citizens, whether products, resources or ideas” [5]. The traditional belief, which holds the view that population size lowers per capita income and sees income as a stock, may be inaccurate. According to Gao [47], on the one hand, regions with high per capita income foster strong attraction for migrants, leading to population growth; on the other hand, population size can promote the per capita income of the labor force, and expansion of the population size can significantly improve the income level of the labor force. Therefore, population-size and environment-quality indices are introduced to construct an analytical framework. For an N-type EKC, the environmental quality (EQ) will show a declining trend with the expansion of population size in the initial stage; until the population size reaches a certain level, EQ will drop to the lowest point, reflecting the stage of diminishing marginal returns. In the later period, with the scale economies effect caused by population size, the cost of pollutant treatment will decrease, and EQ will increase to the highest point, reflecting the stage of increasing marginal returns. In the terminal stage, the population size will further expand and enter the stage of diseconomies of size. The continuous increase in the total amount of pollutants will lead to increasing treatment costs, and EQ will begin to decline (Figure 2d), re-entering the stage of diminishing marginal returns [48]. When the population size of the study area is between the two inflection points, EQ is in the stage of increasing marginal returns. In other words, as the population size increases, EQ rises. On the contrary, when the population size is on the left of the first inflection point and on the right of the second inflection point, EQ will be in the stage of diminishing marginal returns. The increase in population size will drive the decline in EQ.

(2).Model Calculation

According to the above framework, EQ is used as the explained variable, and population size is used as the explanatory variable. Considering the quadratic and cubic relationship between the two, the formula is as follows:(1)$$EQ_{it}=\alpha +\beta_{1}{n_{it}}^{3}+\beta_{2}{n_{it}}^{2}+\beta_{3}n_{it}+\gamma_{1}GDP+\gamma_{2}UR+\varepsilon_{it}$$
where *i* represents the number of regions (*i* = 1 to 7), namely the seven counties; *t* represents time (*t* = 1 to 10), with the ten years from 2009 to 2018 are selected for analysis; *EQ_it_* is environmental quality; *n_it_* is population size; *GDP* and *UR* are control variables, representing the economic scale and the proportion of urban population, respectively; *ɛ_it_* is the stochastic disturbance satisfying basic assumptions; and *α*, *β*_1_, *β*_2_, *β*_3_, *γ*_1_, and *γ*_2_ are the parameters to be estimated.

As curve shapes may vary between regions, the curve shape can be determined according to the estimated coefficients *β*_1_, *β*_2_, and *β*_3_ in Formula (1); then, the average optimum population size can be calculated.

If *β*_1_ ≠ 0, *β*_2_ ≠ 0, the curve will be the N type. In this case, when *β*_1_ < 0, *β*_2_ > 0, *β*_3_ > *β*_2_^2^*/3β*_1_, the curve will be the inverted N type. When *β*_1_ > 0, *β*_2_ < 0, *β*_3_ < *β*_2_^2^*/3β*_1_, the curve will be the N type.If *β*_1_ = 0, *β*_2_ ≠ 0, the curve will be the U type. In this case, when *β*_2_ > 0, the curve will be the U type. When *β*_2_ < 0, the curve will be the inverted U type.If *β*_1_ = 0, *β*_2_ = 0, the curve will be the linear type.

For an N-type or U-type curve, the first derivative of Formula (1) can be used to obtain the population size at the “inflection point”:(2)$${n_{1,2}}^{*}=\frac{-2\beta_{2}\pm \sqrt{4{\beta_{2}}^{2}-12\beta_{1}\beta_{3}}}{6\beta_{1}}$$
(3)$$n^{*}=-\frac{\beta_{3}}{2\beta_{2}}$$

Formulas (2) and (3) are the population sizes of the “inflection point” corresponding to the N-type and U-type curves, respectively.

(3).Population-Size and Environment-Quality Model Fitting

The data of population size, *GDP*, and the proportion of urban population from 2009 to 2018 were mainly obtained from the *Ningxia Statistical Yearbook (2010–2019)* [49]. The data used to calculate EQ were widely sourced from *Ningxia Statistical Yearbook (2010–2019)* [49] and data obtained from the county governments in previous research, including *Guyuan Yearbook (2017–2019)*, *Xiji Yearbook (2010–2018)*, *Xiji County Economic Information Manual (2010–2018)*, *Longde Yearbook 2006–2017* and *Jingyuan Yearbook (2010–2011)*, *Pengyang Yearbook (2011–2018)*, *Tongxin Yearbook (2013–2018)*, *Yanchi Yearbook (2012–2018)*, *Yanchi County Economic Situation Manual (2009*, *2011*, *2013–2015*, *2017)*, etc. Based on Stata15.1, the fitting was divided into three steps. First, as the pooled model ignores omitted or unobserved heterogeneity and the varying coefficient model ignores the commonality between cross-sectional individuals, the variable intercept model was adopted for regression through the Chow test [50]. Second, the differences of cross-section members needs to be considered because the seven counties have natural and cultural differences, and the impact of time changes also need to be considered because the Chinese government placed considerable importance on the ecological environment of this region from 2009 to 2018 (*National main functional area planning*; *Planning of main functional areas in Ningxia Hui Autonomous Region*; *The 13th five year plan for ecological protection and construction in Ningxia*; et al.). Therefore, a fixed-effect model should be adopted, and a “two-way fixed effect model” should be selected for estimation. Third, according to the Wald test and Breusch-Pagan LM test, the disturbance term has intergroup heteroscedasticity, autocorrelation within panels, and contemporaneous correlation. Therefore, the comprehensive feasible generalized least squares (FGLS) method was selected to estimate the model [51].

#### 4.1.2. Evaluation of Population Carrying Capacity

In this section, we calculated the total available land resources, consumption intensity of land resources, and the upper limit of population size that land resources can carry for each county under the condition of maintaining the current development mode.

(1).Total Available Land Resources

Considering the total size of population that can be supported rather than the size of new population, we referred to the ideas of Niu et al. [4,52] and used the index of total planned construction land to represent available land resources. We obtained the total planned construction land in each county in 2020 from *Ningxia Autonomous Region People’s Government on Xiji County* [53], *Longde County* [54], *Jingyuan County* [55], *Pengyang County* [56], *Tongxin County* [57], *Yanchi County* [58], *Haiyuan County* [59] *Land Use General Plan (adjusted and improved version in 2016) Approval*, although it is not listed due to space limitations.

(2).Consumption Intensity of Land Resources

Compared with developed areas, the change in land resource occupation rates in underdeveloped mountainous areas is relatively small because of the long-term low economic and technological level [60]. Accordingly, the change in land resource consumption intensity in the study area is relatively small. Industrial land area was calculated by dividing the industrial and mining land area in 2017 by the industrial added value in 2017. Per capita living land area was calculated according to the ratio of residential land (urban and rural residential land) in 2017 to permanent population in 2017. The data source of industrial added value and permanent population was the *Ningxia Statistical Yearbook (2018–2019)* [49], and the data on land area was provided by the Ningxia Autonomous Region Department of Natural Resources. The intensity of land use in each county is presented in Table 1.

(3).Population Carrying Capacity of Land Resources

In the evaluation of the population carrying capacity of land resources, the upper limit of population that land resources can carry or accommodate under a given development mode is determined. Industrial and population growth are subject to the restriction of planned construction land. When the carrying capacity of land resources reaches the limit, the total amount of available land resources can be regarded as the sum of industrial land and living land, and the upper limit of population size of each county can be obtained by deducting industrial land use from the total amount of planned construction land [52].
(4)$$pop=\frac{L-IVA\times intenL_{industry}}{intenL_{pop}}$$
where *pop* is the population, *L* represents the total amount of available land resources, *IVA* represents industrial added value, *intenL_industry_* is the industrial land-use intensity, and *intenL_pop_* is per capita residential land area.

#### 4.1.3. Sustainable Population Size

The population size constrained by the “inflection point” of EQ was determined by identifying the collaborative change trend of seven indicators in each county. With the improvement of economic and technological levels in the future, specific indicators may exceed the 2009–2018 levels. Furthermore, under the macro background of county development, the urbanization of Ningxia and even China is in the stage of accelerating rise, and the inflection point may present a certain degree of displacement. Therefore, in the long term, EQ is an elastic constraint of population size with dynamic rather than static characteristics. The population at the “inflection point range” of EQ should be an optimum population, which relies on the assumption that the current EQ does not decline. Limited by the supply of natural resources and the increasing difficulty of marginal utilization [61], land resources are rigid constraints of population size and belong to the category of population carrying capacity. According to the principle of combining elasticity and rigidity, the regional sustainable population size should be determined with the lower limit of no degradation of EQ and the upper limit of land resource carrying capacity. The specific criteria are as follows:1.The current population size is lower than the upper limit of land resource carrying capacity;If the population is increased within the upper limit, EQ will decline, that is, the marginal returns of EQ will diminish. We suggest that the current level of sustainable population size be maintained;If the increase in population within the upper limit improves EQ, that is, the marginal returns of EQ increase, we suggest that the sustainable population size be maintained between the current situation and the upper limit and within the “inflection point” of EQ.2.The current population size is higher than the upper limit of land resource carrying capacity;Therefore, we recommended that the sustainable population size be reduced to the upper-limit level.

### 4.2. Variable Selection and Data Processing

#### 4.2.1. Variable Selection

EQ is a comprehensive index, and it was divided into three categories and seven indicators for analysis (Table 2) by considering the availability of county data and referring to Xu [48]. Considering that the seven counties are underdeveloped mountainous areas, traditional industries, extractive industries, and domestic pollutant emissions are the main sources of the effect of population on the environment. Four indices, comprising the “volume of sulfur dioxide emission per square kilometer” (PL), the “volume of industrial soot (dust) emission per square kilometer” (PY), the “volume of industrial nitrogen oxides discharged per square kilometer” (PD), and the “annual proportion of excellent days in terms of air quality” (KQ), were selected as indicators of the degree of environmental pollution. The seven counties, mainly distributed around the foot of Mount Liupan, belong to the farming–pastoral zone of northwestern China and lie in the transitional eco-sensitive zones of the forest and dry grassland of the Loess Plateau. In particular, they are restricted ecological zones in China. The indicator “afforestation area this year” (ZL) was selected to reflect the maintenance level of ecological products, and “forest and grass coverage rate” (LC) was selected to represent the overall service level of the regional ecological base to establish the ecosystem foundation. As the core index, “total energy consumption by GDP” (NH) reflects the resource consumption level influenced by EQ attributable to the fossil fuel economy and represents the energy consumption intensity of economic activities. Population size is expressed by the resident population of each county and is the dependent variable of the EQ index.

#### 4.2.2. Data Sources and Processing

As the county-level government does not calculate the comprehensive EQ index, seven indicators were comprehensively measured in this study. At present, there are many calculation methods of multi-index composite indices; here, we present two representative methods. The first method involves the determination of a comprehensive value through nondimensionalization and weighting, although the calculation result is highly dependent on the selection of weights and indices, with certain subjectivity [62]. The second is principal component analysis (PCA), in which dimension reduction is applied to eliminate repetitive or closely related original variables, and a group of new, unrelated, comprehensive variables is generated while retaining the original information as much as possible. PCA can simplify the index structure and fundamentally solve the problem of information overlap between variables [63]. Compared with classical principal component analysis, the global principal component analysis (GPCA) method combines PCA and the time series analysis method; the GPCA method can ensure the unity, integrity, and comparability of system analysis. Accordingly, we used the GPCA method to construct an environment quality index for the seven counties in Xihaigu Mountain area from 2009 to 2018.

We used the Kaiser–Meyer–Olkin (KMO) test and Bartlett’s test to determine whether the data can be analyzed using the GPCA method. The result of the KMO test is 0.710 (>0.5), which indicates that there is a strong correlation among test indicators. The approximate chi-square of Bartlett’s test is 89.566, and the significance level is 0.000 (<0.01), indicating that the result rejects the null hypothesis. Therefore, the data can be analyzed using the GPCA method. Specifically, the eigenvalue method was used, and multiple indicators were fitted to EQ through SPSS25; finally, three principal components were determined on the premise that the eigenvalue is greater than 1.

Among them, specific data on the following parameters at individual points are lacking: PL of each county from 2010 to 2011 and PY and PD of each county from 2009 to 2011. NH of each county in 2009 and 2010 is also lacking. Moreover, KQ of Haiyuan has been announced since 2012, whereas that of other counties has been announced since 2013. Nevertheless, regular results were obtained on the macro level, and the missing data (calculated on average) had little influence on the results [64].

In addition to the GPCA of the whole, we also carried out temporal principal component analysis for each county to explore the driving factors of environmental quality in each county, with the component matrix shown in Table 3. According to the principal component matrix of the EQ index system of the seven counties, the load coefficient of each index greater than 0.5 was not concentrated in the principal components, and the characteristics of each principal component were not prominent, indicating that the contribution of all selected indices to EQ has low repeatability and that the selected indices were reasonably distributed.

We obtained score sequences corresponding to three principal components in the process of GPCA. After testing, the scoring sequences of the second and third principal components were poorly fitted to the population size, so no further analysis was performed. In the subsequent analysis, the first principal component was used as the comprehensive EQ, that is, we took the score sequence of the first principal component as an EQ indicator and used its regression analysis with population size to explore the relationship between population size and EQ.

## 5. Results

### 5.1. Population Size under the Constraints of Environmental Quality

(1).Inverted N-Shaped Curve and Population Inflection Point

Population-size and environment-quality model estimation results show that the population size and EQ conform to an inverted N-shaped curve (Figure 3). In the results, all variables except GDP are significant, and the cubic coefficient is significantly non-zero according to the Wald test (Table 4). The parameter estimation results show that UR is positively correlated with environmental quality. The reason for this phenomenon may be that urbanization has changed original production methods and lifestyles, making energy use more intensive. Moreover, the technology spillover effect caused by the agglomeration of factors also enables the rapid diffusion of emission reduction technologies within cities and towns so as to achieve improved emission reduction effects. The parameter estimation results also show that GDP has no significant impact on environmental quality, which indicates that the relationship between the two is not a simple linear relationship but may be a more complex quadratic or even cubic relationship, as indicated by the N-type EKC curve mentioned earlier in this paper. However, we mainly obtained the relationship between environmental quality and population, whereas GDP was used as a control variable and will not be analyzed in detail for the time being.

According to Formula (2) and Table 4, the average population sizes of the seven counties at two inflection points were 264,300 and 416,900, respectively. When the population size is less than 264,300, EQ and population size are inversely related, that is, EQ decreases with increased population size. The lowest EQ occurs when the population size is close to the inflection point. When the population size is between 264,300 and 416,900, EQ changes in the same direction as the population size, that is, EQ increases with increasing population size and reaches the “inflection point” again when the population size reaches 416,900. After this point, EQ deteriorates continuously with increasing population size.

(2).Driving Factors of Environmental Quality Change

Specific indicators constrain EQ and thus affect the sustainable population size. According to the factor loading matrix (Table 3), each index plays different roles depending on the county in principal component 1. Among them, PL, PY, PD, LC, and NH have significant effects on principal component 1 in Xiji and Jingyuan, whereas the impact of ZL is relatively weak. This indicates that the degree of environmental pollution and the energy consumption intensity of economic activities in Xiji and Jingyuan pose a prominent threat to EQ. In Longde and Pengyang, indicators of relatively higher significance were PL, KQ, ZL, and LC, which are mainly concentrated in the environmental pollution and ecological system aspects, whereas PY and PD have a relatively weak influence. The Loess Hills are widely distributed in Pengyang and Longde, and Pengyang hosts large-scale coal mining, which has led to the prominent influence of ecological indicators. The main influencing factors in Haiyuan and Tongxin are PL, PY, and PD. NH also has a significant impact on Tongxin. The high emissions, high energy consumption, and low-output production models of some enterprises in the two counties impose increased pressure on EQ. The polluting industries in Tongxin include food processing of agricultural products (wolfberry, grape, etc.) and the manufacture of leather, fur, feather, and related products (cashmere), which are included in the “top ten polluting industries”. The polluting industries in Haiyuan include food processing of agricultural products (beef cattle, etc.). In Yanchi, PY, PD, and NH have a considerable impact on air quality, which can be mainly attributed to Gaoshawo Industrial Park, i.e., coal and petroleum processing. Overall, the seven counties showed differences as well as similarities. The key driving factors of EQ change over the entire region include PY, PD, LC, and NH.

(3).Environmental Economic Stage of the Counties’ Current Population Size

According to the seventh National Population Census data of China, in 2020, Xiji, Tongxin, and Haiyuan had populations of 315,800, 320,800, and 333,500, respectively, all surpassing the lowest inflection point of EQ deterioration, placing them in the stage of increasing marginal returns, although their EQ values are relatively poor compared to those of the four other counties. Jingyuan has the highest EQ and ranked along the curve of diminishing marginal returns, together with Longde, Yanchi, and Pengyang. Among the seven counties, Haiyuan is in the middle and late stage of increasing marginal returns, which is closest to the highest point of EQ improvement. With the continuous increase in population size in the future, it may enter the stage of diminishing returns. On the whole, there was no “inflection point” of the lowest EQ in the limited development ecological zones of the Xihaigu Mountain area, which may be related to economic underdevelopment, especially the superposition of the total population change and ecological migration in the counties.

On the whole, except for Tongxin, the population size of all other counties was relatively high in 2005 and plunged in 2010 before re-entering a state of growth (Figure 4). Yang et al. [65] showed that from 2001 to 2009, there were 560,000 ecological immigrants in the seven counties, which significantly improved the ecological environment of the emigrated areas, although it significantly decreased in 2010. Since then, population growth in each county has slowed down and picked up again. In particular, population growth in Tongxin, Jingyuan, Haiyuan, and Xiji has rebounded significantly, and their national minority population ratios ranked first, second, third, and sixth, respectively, among 22 counties (districts) in the whole region. This situation may be considerably influenced by the fertility concept is highly correlated with natural population growth (Table 5). According to *Ningxia Statistical Yearbook (2019)*, in 2018, the natural population growth rate of five of the seven counties was more than 10%, which is much higher than the average level of 7.78‰ in Ningxia. The rapid growth of permanent population in Yanchi in the later period is mainly due to its proximity to the provincial capital of Yinchuan and the energy and chemical industry base of Ningdong, as well as the large amount of mechanical migration caused by the relatively developed petroleum industry and agricultural industry [66]. Furthermore, from the perspective of the population-size and environment-quality models, with the exception of Tongxin and Xiji, a contradiction between risk of population size growth and EQ decline was observed in all other counties.

### 5.2. Population Size under the Constraints of Land Resources

Under the constraints of land resources (Figure 5), Jingyuan and Haiyuan can accommodate the least (137,300) and most (487,500) population, respectively, showing a difference of more than 350,000 people. Longde (224,700), Pengyang (203,200), and Yanchi (200,000) are at the same level, with smaller supporting populations. Xiji (425,100) and Tongxin (379,800) have a relatively high population size. Yanchi and Pengyang are the most severely constrained counties by land resources, with populations of 159,200 and 160,500, respectively, in the 2020 census. However, as their land resource carrying capacities are only 200,000 and 203,200, respectively, under the current development mode, the population may exceed the carrying capacity if growth continues. In contrast, Tongxin county and Jingyuan county are subject to fewer land constraints and can support additional populations of 59,200 and 51,600, respectively, under the current development mode. Haiyuan, Longde, and Xiji are the least restricted by land resources, with a surplus carrying capacity of 154,000, 115,200, and 109,300, respectively.

### 5.3. Sustainable Population Size

In this study, we integrated the population sizes constrained by land resource carrying capacity and EQ (Figure 5). The land resource carrying capacities of Jingyuan, Pengyang, Yanchi, and Longde are similar to the population size in 2020, and all of them are located to the left of the first inflection point. This implies that under the condition of maintaining the current industrial development mode, they can only be in the stage of diminishing marginal returns, and the sustainable population size corresponds to the current level. The current population of Tongxin is in the stage of increasing marginal returns, and an appropriate increase in population size will improve EQ. When it is close to the maximum land resource carrying capacity, EQ will reach the optimal level, and the sustainable population size will range from approximately 320,800 to 379,800. The marginal incomes of the current population of Xiji and Haiyuan are all in the increasing stage, indicating that an appropriate increase in population size will improve environmental quality, reaching the optimal level as the population size approaches the second inflection point. However, if the population sizes of these two counties cross the second turning point, even if there are remaining land resources, it is not appropriate to continue to increase the population; otherwise, the environmental quality will decline. Above all, the sustainable population sizes of Xiji and Haiyuan are 315,800–416,900 and 333,500–416,900, respectively.

## 6. Discussion

### 6.1. Necessity of Sustainable Population Size Research

Sustainable population size can be determined by superimposing the current population, optimum population, and population carrying capacity. Some scholars doubt the objectivity and research value of investigating population capacity or population carrying capacity, although most scholars hold a positive attitude and believe that relevant research is highly significant [67]. Pan [68] believes that although population capacity is a reference value in a certain sense, its role in population policy, population planning, and national economic development planning is undeniable. Shen [69] points out that the contradiction between land and population is prominent in China and may become a long-term constraint factor affecting China’s economic development. Research on land population carrying capacity is the fundamental basis for formulating coordinated development of population–land policies. In addition, although the seventh survey results show that the national population growth rate is slowing down, the fact that the total population and economic activities of 1.4 billion people have exerted considerable pressure on the ecological environment remains indisputable. Production and consumption activities associated with the livelihood of populations are responsible for the pressure on the ecological environment. Therefore, even if urbanization is accelerated and the growth rate slows down in the future, the probability of ecological environmental overload induced by the expansion of economic scale and improvement of quality of life should be taken into account. Furthermore, the implementation of the three-child policy in China will have varying effects in less developed areas compared to the national scale, and its effect is likely to be prominent. The fact that the population growth rate is higher in Tongxin, Haiyuan, Xiji, and Jingyuan, with higher proportions of minority population, shows the important role of cultural characteristics and ideas and highlights the necessity of research. Furthermore, it reveals that future population policies for less developed regions should not simply be based on the macroscopic results of the population size of the whole country. In other words, a “one size fits all” approach should be avoided.

### 6.2. Dynamics of Sustainable Population Size Accounting

The quantitative analysis of population using the proposed model is subject to a certain “abstractness” limitation. However, it is difficult to solve the problem through qualitative interpretation alone (without a model). The model can provide results that cannot be obtained through general qualitative reasoning, which can improve the operability of research [70]. “Resources” provide humans with tangible means of production and a material basis for production and life, whereas the “environment” provides life support, waste absorption, and other functions; both are necessary conditions for human survival and development. The evaluation of land resource carrying capacity is based on the perspective of resources, which corresponds to rigid constraints and focuses on population carrying capacity. The population-size and environment-quality model was developed with consideration of the environmental perspective, which corresponds to elastic constraints and focuses on the optimum population. In China, current studies mainly focus on the status quo, static analysis, and evaluation or only on resources and the environment, or considering the rigid bearing capacity of both, largely neglecting regional dynamic change processes and predictions of development trends, with overall “dynamic” research [71]. Considering the current population, rigid constraints, and elastic constraints, the results of the three analyses were superimposed to ascertain a dynamic range of sustainable population sizes, which overcomes the shortcoming of only considering the maximum carrying size to a limited extent. The findings have certain reference significance for the implementation of regional sustainable development measures.

### 6.3. Comparative Discussion with the Existing Research

Compared with existing research that mostly focuses on the relationship between economy and environment, we built a model of the relationship between environmental quality and population that reflects the similar relationship with EKC. We adopted the method of integration and superposition to obtain the sustainable population size as a whole. In other words, after obtaining the results from an environmental point of view, we again obtained the population size by using the two indicators of industrial land use intensity and per capita living land area. In this way, economic indicators are considered separately, and the economy and population size are linked through land resource constraints. The energy consumption per unit of GDP in each county can also indirectly reflect the relationship between the economy and the environment, although we did not analyze the economic indicators in detail because we mainly obtained the relationship between environmental quality and population. By using the data of counties in Ningxia restricted development ecological zone from 2009 to 2018, we found that there is a close relationship between population size and environmental quality in Xihaigu Mountain area, which is consistent with the views of many scholars [38,72,73,74,75,76], although most studies have only explored the positive or negative effects of population size on environmental indicators. More specifically, we introduced the square and cubic terms of the population size and found that the environmental quality and population size in Xihaigu area showed an inverted N-shaped curve. Our results are consistent with Xu’s [48] view that the relationship between population size and environmental quality should show an inverted N-shaped curve, although his empirical results show a positive N-shaped curve due to the particularity of urbanization and insufficient selection of indicators. In addition, this we evaluated the population carrying capacity of counties in Xihaigu Mountain area from the perspective of total available land resources. The results show that although the carrying capacity was low, all counties were not overloaded, which is consistent with the research results of other scholars on Ningxia [77,78]. Yanchi County and Pengyang County are particularly constrained by land resources.

## 7. Conclusions and Policy Implications

In this study, we investigated the sustainable population size of underdeveloped areas at the county level. To this end, seven counties in limited development ecological zones of the Xihaigu Mountain area in Ningxia were selected as the research object. We discussed the sustainable population size of each county from the bidirectional perspective of environment and resources. The following main conclusions can be drawn.

First, from 2009 to 2018, the relationship between EQ and population size of the seven counties showed an inverted N-shaped change, which supports our hypothesis of an N-shaped population-size and environment-quality model. The theoretical inflection point values of the average population size of each county were 264,300 and 416,900. The population size was mainly restricted by the volume of industrial soot (dust) emissions, the volume of discharged industrial nitrogen oxides, the forest and grass coverage rate, and total energy consumption by GDP. According to the situation in 2020, with the exceptions of Xiji, Tongxin, and Haiyuan, the other four counties had population sizes with relatively good EQ, which may be related to ecological migration. Although the external intervention of immigration policies has considerably relieved the pressure on local population, it has not fundamentally solved the ecological environment problem of population growth. Except for Xiji, Tongxin, and Haiyuan, the population size and carrying capacity of the other counties are in the stage of diminishing marginal returns of EQ, and there is a contradiction between population size growth and EQ decline.

Second, restricted by land resources, Jingyuan has the lowest population capacity, followed by Longde, Pengyang, and Yanchi. Xiji and Tongxin have relatively high capacities, and Haiyuan has the highest capacity. Jingyuan, with the lowest capacity, which can be attributed to its natural geographical conditions; Jingyuan also had the smallest planned construction land area in 2020. Although the planned construction land area of Longde is also small, its industrial land use intensity is much lower than that of Jingyuan, leading to a larger population carrying capacity of land. The most severely restricted counties in terms of land resources are Yanchi and Pengyang. Under the condition of maintaining the current development mode, they can carry 40,800 people and 42,700 people, respectively, with the risk of exceeding the carrying capacity. In contrast, Tongxin and Jingyuan are less constrained by land resources, and Haiyuan, Longde, and Xiji are the least constrained by land resources.

Third, to maintain sustainable development, the population sizes of Tongxin, Xiji, and Haiyuan should be in the range of 320,800–379,800, 315,800–416,900, and 333,500–416,900, respectively. As for Jingyuan, Pengyang, Yanchi, and Longde, their current scales of 85,000, 160,500, 159,200, and 109,500, respectively, should be maintained. In the future, during the implementation of the main function zone strategy, surplus construction land should be used for the development of environmentally friendly industries rather than for population expansion. In addition to ecological migration and urbanization, it is more important to gradually change the fertility mindset of local residents. Moreover, the pressure of local population should be alleviated through voluntary population migration to improve living standards.

Fourth, in terms of the methodology, in this study, we initially analyzed the current population size of counties and continued the analysis with according to the concept that the elastic constraint of environmental quality determines the bottom line, whereas rigid constraints of land resources determine the upper limit. This two-way perspective not only takes into account the optimum population but also superimposes the maximum carrying capacity. In this manner, the population size can be obtained within the bounds of the upper and lower limits. To some extent, it overcomes the shortcoming of stasis in traditional methods of resource and environmental carrying capacity. The research perspective and calculation method have a certain informative and complementary role in enriching scientific research on sustainability.

Based on the above conclusions, we put forward three key policy implications.

First, the control of pollution emissions and energy consumption should be considered. According to this study, the volume of industrial soot (dust) emissions, the volume of discharged industrial nitrogen oxides, and total energy consumption by GDP are obvious indicators that restrict the population size of the study area. The government should focus on high-emission enterprises and improve the level and ability of pollution control. Its implications lie in the development of the ecological industry, environmentally friendly industry, and even concentrated development of green agriculture, which are important directions for the future.

Second, the population size should be controlled within a reasonable range. The inverted N-shaped fitting results of the population-size and environment-quality model prove the importance of multiple population policies in terms of environmental quality in long run. To maintain sustainable development, the population size of Tongxin, Xiji, and Haiyuan should be in the range of 320,800–379,800, 315,800–416,900, and 333,500–416,900, respectively. As for Jingyuan, Pengyang, Yanchi, and Longde, their current population sizes of 85,000, 160,500, 159,200, and 109,500, respectively, should be maintained. It should be emphasized that the population size discussed here is not absolute but a relative value of the permanent population, considering the openness of the region. Moreover, the aim of controlling the population size is not to curb the increase in the number of births. Migration arguably plays an increasingly important role in regional growth and development than ever before [79]. On the basis of ecological migration policy, population control can be achieved through spontaneous population flow, such by as urbanization. Through dynamic and static comparative analysis of production and population distribution, Li and Fan [80] found that the polarization effect of production in China was not high and that the polarization effect of population was extremely low; therefore, they encouraged the impoverished population in central and western China to move to the east, especially to the eastern core regions.

Third, cultural characteristics and concepts should be considered, differentiated population size control policies should be developed. Although the external intervention of immigration policy was effective and considerably alleviated the pressure on local population, it could not fundamentally solve the ecological and environmental problems of population growth. The higher population growth rates in Tongxin, Haiyuan, Xiji, and Jingyuan, with higher proportions of minority population, indicate the important role of cultural characteristics and ideas and highlight the necessity of research. Furthermore, these growth rates reveal that future population policies for less developed regions should not simply be based on the macroscopic results of the population size of the whole country. In other words, a “one size fits all” approach should be avoided.

In this paper, we explored the county-level sustainable population size in Xihaigu Mountain area and drew some conclusions; however, some limitations should be considered. Due to the lack of standardized statistical data in Chinese counties, the population carrying capacities of other resources, such as water, were not evaluated in this paper. In addition, the model may be overly simple, although it is still of value for scholars to discuss, and we will continue to study and explore a more perfect framework. Future research can better refine the environmental indicators and resource types to obtain a more accurate range. Finally, apart from population size, attention should also be paid to population structure and population quality in the future.

## Figures and Tables

**Figure 1 ijerph-19-09560-f001:**
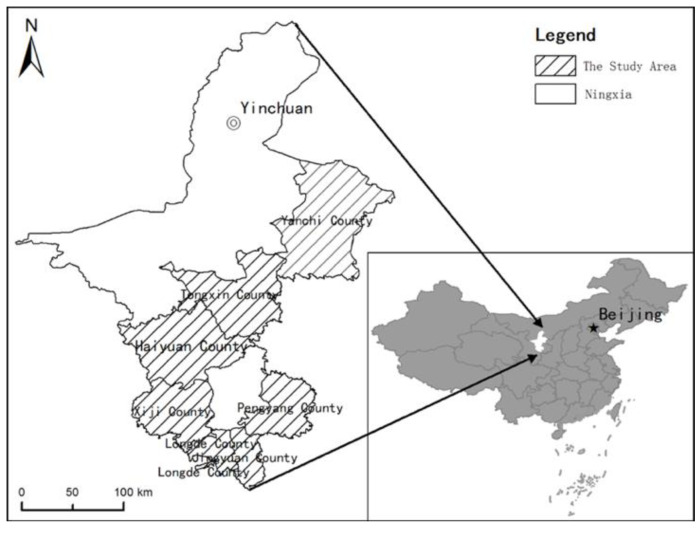
Location of the research area.

**Figure 2 ijerph-19-09560-f002:**
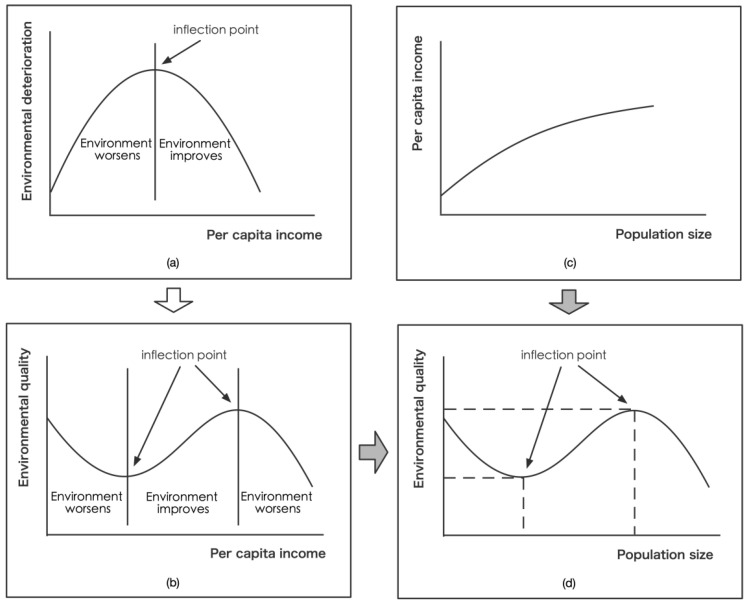
Derivation diagram of the relationship between population size and environmental quality: (**a**) environmental Kuznets curve; (**b**) graph of the relationship between population size and per capita income; (**c**) long-term relationship between per capita income and environmental quality; (**d**) diagram of the relationship between population size and environmental quality.

**Figure 3 ijerph-19-09560-f003:**
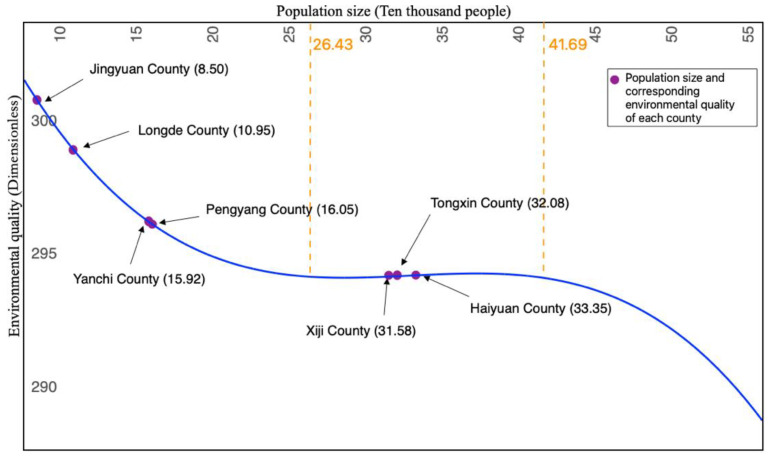
Analysis chart of the population-size and environment-quality model, as well as the current situation. The data in brackets indicate the population of each county in the seventh National Population Census of China (2020).

**Figure 4 ijerph-19-09560-f004:**
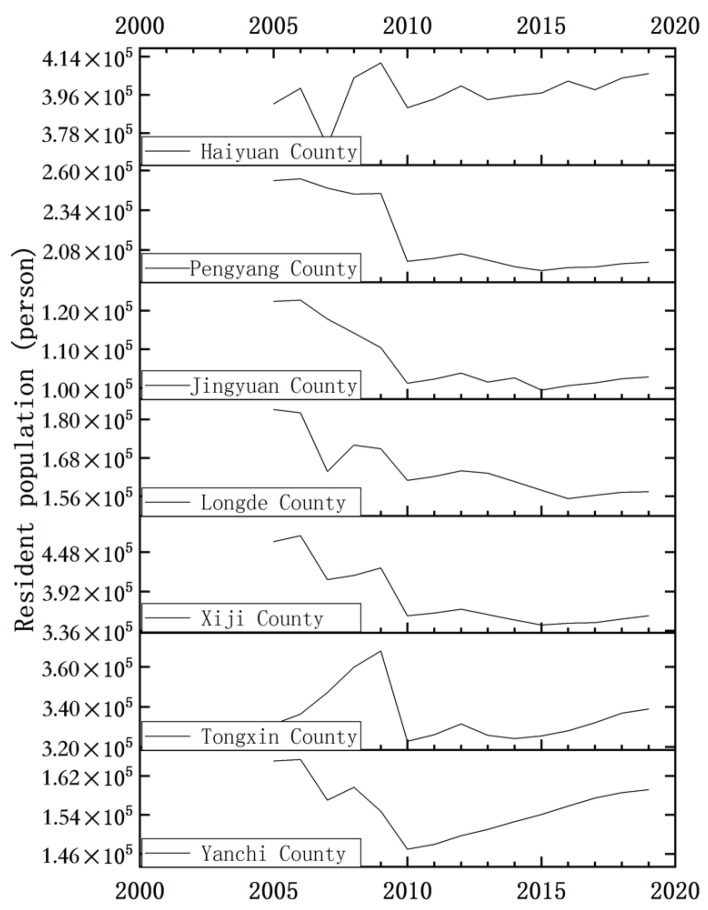
Chart of changes in resident population of each county in the study area from 2005 to 2019.

**Figure 5 ijerph-19-09560-f005:**
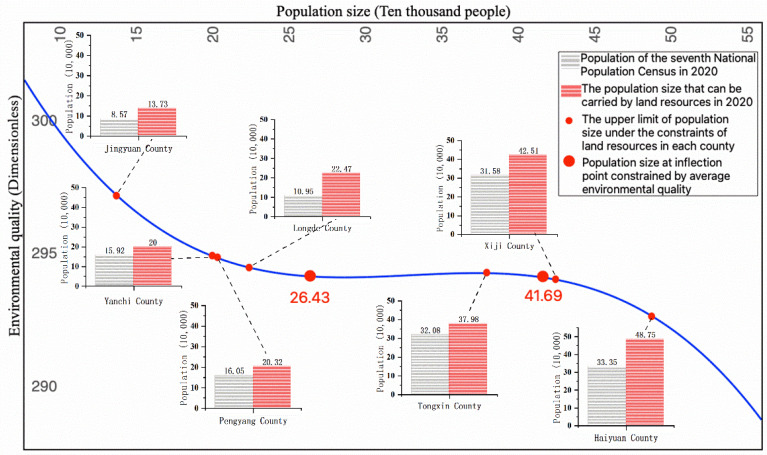
Interval analysis of sustainable population size at the county level in the limited development ecological zones of Ningxia.

**Table 1 ijerph-19-09560-t001:** Industrial and residential land use intensity in each county.

Administrative Region	Industrial Land Use Intensity (ha/100 Million Yuan)	Per Capita Living Land Area (m^2^/Person)
Xiji county	45.85	383.38
Longde County	24.83	264.14
Jingyuan County	64.64	313.53
Pengyang County	27.35	570.08
Tongxin County	21.33	430.99
Yanchi County	36.63	888.87
Haiyuan County	62.36	385.39

**Table 2 ijerph-19-09560-t002:** Variables and related indicators.

Variable	Category	Indicators	Unit
Population size (n)		Total population	Ten thousand people
Environmental quality (EQ)	Degree of environmental pollution	Volume of sulfur dioxide emissions per square kilometer (PL)	t/100 hm^2^
		Volume of industrial soot (dust) emissions per square kilometer (PY)	t/100 hm^2^
		Volume of industrial nitrogen oxides discharged per square kilometer (PD)	t/100 hm^2^
		Annual proportion of excellent days of air quality (KQ)	%
	Ecosystem foundation	Afforestation area this year (ZL)	khm^2^
		Forest and grass coverage rate (LC)	%
	Energy consumption intensity of economic activities	Total energy consumption by GDP (NH)	tce/10,000 Yuan

**Table 3 ijerph-19-09560-t003:** Principal component matrix of the environmental quality index system of each administrative region.

Administrative Region	Principal Component	PL	PY	PD	KQ	ZL	LC	NH
Seven Counties	1	0.624	0.697	0.890	0.166	−0.501	0.755	0.650
	2	−0.545	−0.300	0.265	−0.200	0.566	0.331	0.585
	3	−0.098	−0.302	0.051	0.934	0.037	0.207	−0.013
Xiji County	1	0.709	0.656	0.87	0.375	−0.069	−0.721	0.939
	2	0.444	−0.427	0.198	0.577	0.851	0.452	−0.042
	3	0.495	0.487	−0.159	−0.682	0.437	0.072	−0.206
Longde County	1	0.851	0.038	0.548	0.635	0.664	0.641	−0.564
	2	0.094	0.846	0.727	−0.493	−0.472	0.08	−0.115
	3	0.461	0.421	−0.16	0.508	−0.112	−0.597	0.328
Pengyang County	1	0.497	−0.403	−0.029	0.872	0.836	0.879	−0.563
	2	−0.301	0.641	−0.716	0.357	0.469	0.043	0.629
	3	0.757	0.463	0.638	0.081	0.152	−0.123	0.465
Jingyuan County	1	0.915	0.521	0.617	−0.869	0.273	−0.787	0.635
	2	−0.344	0.591	0.698	0.45	0.086	0.578	0.628
	3	−0.084	−0.449	−0.057	0.152	0.922	−0.069	0.269
Haiyuan County	1	0.93	0.699	0.591	−0.585	−0.624	0.636	−0.06
	2	0.283	−0.585	0.23	0.754	−0.212	0.424	−0.808
	3	0.01	0.341	0.729	0.003	0.593	−0.527	−0.465
Tongxin County	1	0.634	0.929	0.909	−0.34	−0.534	0.002	−0.82
	2	−0.385	0.207	0.173	−0.476	0.622	0.978	−0.076
	3	0.575	0.077	−0.252	−0.608	0.109	−0.077	0.434
Yanchi County	1	−0.215	0.771	0.934	0.39	0.123	−0.549	0.805
	2	0.886	−0.197	0.257	−0.184	0.772	0.52	0.454
	3	0.342	0.44	0.087	0.549	−0.493	0.632	−0.19

**Table 4 ijerph-19-09560-t004:** Population-size and environmental-quality model estimation results.

Variable	Coef.	Std. Err.	z-Value
n	−1.5988 ***	0.4953	−3.23
n^2^	0.0494 ***	0.1700	2.91
n^3^	−0.0005 ***	0.0002	−2.65
GDP	0.0015	0.0060	0.24
UR	0.0379 ***	0.0058	6.51
Constant	311.0892 ***	68.4265	4.55
chi2(1)	7.02 ***	-	-

*** *p* < 0.01.

**Table 5 ijerph-19-09560-t005:** Population situation of each county in the study area from 2009 to 2018.

Administrative Region	Average Natural Growth Rate (‰)	Average Proportion of Ethnic Minority Population (%)	Ranking of the Proportion of Minority Population *
Xiji County	12.46	57.63	6
Longde County	6.57	11.99	19
Pengyang County	9.92	30.42	10
Jingyuan County	10.63	78.99	2
Haiyuan County	12.94	71.65	3
Tongxin County	13.08	87.58	1
Yanchi County	8.79	2.32	22
Ningxia	8.73	35.83	

* Proportion of ethnic minority population in each county, ranked among 22 counties (districts) in Ningxia.

## Data Availability

Data supporting the reported results are available upon request to the first author.

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
