# Peer review of "Sustainable Population Size at the County Level under Limited Development Policy Constraints: Case Study of the Xihaigu Mountain Area, Northwest China"

_ijerph, 2022, doi:10.3390/ijerph19159560_

Round 1

Reviewer 1 Report

This study analyzes the impact of population size on environmental quality in the Xihaigu Mountain Area, China, and derives an appropriate population size. Specifically, it shows that the environmental quality relative to the population is an inverted N-shaped curve, and that the range of inflection points is the range of appropriate population size. It is understandable that this is a highly necessary research theme in China, where environmental protection is required in the midst of development. However, there are fatal errors in the analysis, and the results are not reliable. For this reason, we judge it to be rejected.

Specifically, an inverse N-shaped curve is estimated from the relationship between the environmental quality index and the population; the environmental quality index is obtained by principal component analysis for each county. Different principal component analyses were performed for each County, and they are not inherently comparable across Counties. In other words, the indicators are integrated based on different weights, resulting in environmental indicators with different criteria. Even if we compare them side by side, we cannot compare the quality of the environment based on the value of indices. Therefore, the meaning of the obtained inverted N-shaped curve is ambiguous, and the interpretation of the analysis results based on this curve is incorrect.

The above are the reasons for rejection, but there are other unreliable points.

The inflection point on the right side of the inverted N-shaped curve in Figure 3 is outside the range of the data and is extrapolated. Statistical indicators are needed to determine how reliable the cubic function obtained is. It is necessary to show at least the t-value or z-value of the parameters obtained, the p-value of the regression equation, and other basic statistical indicators. Also, I believe the data was collected for the decade 2009-2018, but Figure 3 appears to show only one point per County. An explanation is needed as to why not all data were used.

Equation (1) also shows that GDP and UR are also explanatory variables, but their parameters are not estimated. While the EKC and N-shaped curves presented in Method assume that economic growth improves the environment by increasing the ability to bear the cost of environmental technologies, this study does not explicitly analyze the effect of economic growth, but only attempts to evaluate it through regression analysis of population and environmental indicators. Based on the original EKC and N-shaped curve theory, the analysis should reflect economic factors such as GDP.

Also, the equation numbers are incorrect and the explanatory text is confusing as to what it refers to. Careful review is required before submission.

Reviewer 2 Report

The detailed comments are as follows:

1.      The title of the work emphasizes the limitations of development policy, although there is no reference to this in the goals set.

2.      There is no clear indication in the introduction before the specifics of Xihaigu Mountain (line 60) as a research area that the research is about just this one example.

3.      The description of the research area (3. Study area) lacks basic information about the changes in the number of people in the past and about the size of the basic demographic indicators determining the number of people, i.e. population growth, etc.

4.      In the methodological part, I lack general information on what data sources were used, e.g. public statistics or own research. Detailed literature items (unfortunately without reference to the relevant items from the references list) are indicated in section 4.2.2, but this way in my opinion is relatively unreadable and I would suggest adding general information about the data sources used earlier in the methodology.

5.      There is a discrepancy between the research area suggested in the title (one area) and the indications in the specific objectives where reference is made to other counties (see comment below).

In general, the problem is for the international reader to understand what is the spatial scope of the research, due to the lack of its proper description. The administrative division of the research area and the relationship between the title "area" and the further analyzed areas should be explained in detail - Xihaigu Mountain Area appears in the title, other areas are further analyzed (objectives -lines 107-109 refer to several counties), in the empirical part there is "Ningxia".

Reviewer 3 Report

This article appears to be well organized and structured and uses appropriate methodologies to explore the computation of the dynamic range of population size under dual constraints of resource and environment. I didn't notice any errors in the text, figures, or tables. However, several errors need to be corrected:

In line 36, list the literature by which Daly proposed principles and concepts, and perhaps also for the Chow test (Gregory C) in line 269.

In lines 308 and 419, the formulas have mark 1 again, and in line 420 there is a link to formula (5), which is not there. In line 460, the other five counties are described, when three are already mentioned, and only 7 counties are analyzed.

Round 2

Reviewer 1 Report

The author does not seem to understand the meaning of the previous peer review comments. Principal component scores represent synthetic variables based on the variance information of multivariate data, and different principal component loadings, which are synthetic weights, will result in different scores even for exactly the same city conditions. Although it should not make sense to compare those principal component scores with different principal component loadings across cities, the authors argue that they are comparable because the original variables used are the same. However, it is theoretically incorrect to compare them and discuss whether the environment is good or bad, since different weights of the same variable produce different results.
